# Co-Clinical Imaging Metadata Information (CIMI) for Cancer Research to Promote Open Science, Standardization, and Reproducibility in Preclinical Imaging

Stephen M. Moore [1,*], James D. Quirk [1], Andrew W. Lassiter [1], Richard Laforest [1], Gregory D. Ayers [2], Cristian T. Badea [3], Andriy Y. Fedorov [4], Paul E. Kinahan [5], Matthew Holbrook [3], Peder E. Z. Larson [6], Renuka Sriram [6], Thomas L. Chenevert [7], Dariya Malyarenko [7], John Kurhanewicz [6], A. McGarry Houghton [8], Brian D. Ross [7], Stephen Pickup [9], James C. Gee [9], Rong Zhou [9], Seth T. Gammon [10], Henry Charles Manning [10], Raheleh Roudi [11], Heike E. Daldrup-Link [11], Michael T. Lewis [12], Daniel L. Rubin [13], Thomas E. Yankeelov [14,15] and Kooresh I. Shoghi [16,*]

1  Mallinckrodt Institute of Radiology, Washington University School of Medicine, St. Louis, MO 63110, USA
2  Department of Biostatistics, Vanderbilt University, Nashville, TN 37235, USA
3  Quantitative Imaging and Analysis Lab, Department of Radiology, Duke University, Durham, NC 27708, USA
4  Brigham and Women's Hospital and Harvard Medical School, Boston, MA 02115, USA
5  Department of Radiology, University of Washington, Seattle, WA 98195, USA
6  Department of Radiology and Biomedical Imaging, University of California, San Francisco, CA 94143, USA
7  Department of Radiology, University of Michigan Medical School, Ann Arbor, MI 48109, USA
8  Fred Hutchinson Cancer Center, Seattle, WA 98109, USA
9  Department of Radiology, Abramson Cancer Center, University of Pennsylvania, Philadelphia, PA 19104, USA
10 Department of Cancer Systems Imaging, Division of Diagnostic Imaging, The University of Texas MD Anderson Cancer Center, Houston, TX 77030, USA
11 Department of Radiology, Stanford University School of Medicine, Stanford, CA 94305, USA
12 Dan L Duncan Comprehensive Cancer Center, Departments of Molecular and Cellular Biology and Radiology, Baylor College of Medicine, Houston, TX 77030, USA
13 Departments of Biomedical Data Science, Radiology and Medicine, Stanford University School of Medicine, Stanford, CA 94305, USA
14 Departments of Biomedical Engineering, Diagnostic Medicine and Oncology, Oden Institute for Computational and Engineering Sciences, Livestrong Cancer Institutes, The University of Texas at Austin, Austin, TX 78712, USA
15 Department of Imaging Physics, The University of Texas MD Anderson Cancer Center, Houston, TX 77030, USA
16 Mallinckrodt Institute of Radiology, Department of Biomedical Engineering, Siteman Cancer Center, Washington University School of Medicine, St. Louis, MO 63110, USA
*  Correspondence: moore.stephen.m@wustl.edu (S.M.M.); shoghik@wustl.edu (K.I.S.)

**Abstract:** Preclinical imaging is a critical component in translational research with significant complexities in workflow and site differences in deployment. Importantly, the National Cancer Institute's (NCI) precision medicine initiative emphasizes the use of translational co-clinical oncology models to address the biological and molecular bases of cancer prevention and treatment. The use of oncology models, such as patient-derived tumor xenografts (PDX) and genetically engineered mouse models (GEMMs), has ushered in an era of co-clinical trials by which preclinical studies can inform clinical trials and protocols, thus bridging the translational divide in cancer research. Similarly, preclinical imaging fills a translational gap as an enabling technology for translational imaging research. Unlike clinical imaging, where equipment manufacturers strive to meet standards in practice at clinical sites, standards are neither fully developed nor implemented in preclinical imaging. This fundamentally limits the collection and reporting of metadata to qualify preclinical imaging studies, thereby hindering open science and impacting the reproducibility of co-clinical imaging research. To begin to address these issues, the NCI co-clinical imaging research program (CIRP) conducted a survey to identify metadata requirements for reproducible quantitative co-clinical imaging. The enclosed consensus-based report summarizes co-clinical imaging metadata information (CIMI) to support quantitative co-clinical imaging research with broad implications for capturing co-clinical

data, enabling interoperability and data sharing, as well as potentially leading to updates to the preclinical Digital Imaging and Communications in Medicine (DICOM) standard.

**Keywords:** co-clinical imaging; metadata; Digital Imaging and Communications in Medicine (DICOM); preclinical imaging; reproducibility; open science; standardization

## 1. Introduction

Preclinical imaging is increasingly being used in translational cancer research to assess the efficacy of therapeutic regimens, to detect and characterize the heterogeneity of tumors, and to validate imaging biomarkers. Indeed, preclinical imaging instruments parallel those available for clinical imaging [1–3], although with greater emphasis on flexibility for research, leading to significant complexities in the workflow and site differences related to their deployment. However, unlike clinical imaging, there are no standard acquisition protocols nor widely accepted reporting standards in the preclinical imaging domain. Importantly, the National Cancer Institute's (NCI) precision medicine initiative emphasizes the use of translational oncology models to address the biological and molecular bases of cancer prevention and treatment. To that end, numerous academic institutions and commercial entities (JAX, Charles River), as well as the National Cancer Institute (NCI; Patient-Derived Models Repository), have launched wide-ranging patient-derived tumor xenografts (PDX) and genetically engineered mouse model (GEMMs) repositories to support the realization of the precision medicine initiative. The emphasis on PDX and GEMMs has also ushered in the notion of co-clinical trials by which preclinical studies can inform clinical trials [4–7], thus potentially bridging the translational gap in cancer research. With the increased use of PDX and GEMMs in translational preclinical imaging research, there is an increased need to support efforts which enhance the reproducibility of preclinical imaging studies and to promote open science.

A lack of reproducibility in preclinical cancer research, including imaging, has been highlighted by numerous publications [8,9]. Other than promoting open science, data sharing has been suggested as one solution to improve reproducibility. Indeed, the NCI has chosen to establish an open environment in which the oncology community can collaborate to tackle the sundry issues that pertain to the reproducibility of animal model research as required for precision medicine. Prominent among these issues is the transparency of details that document imaging experiments and their application to translational research. Prior efforts, such as the ARRIVE [10] and UKCCCR [11], have highlighted guidelines for reporting animal research. The collection of metadata which captures details regarding the preclinical imaging experiment is critically needed to enhance reproducibility and to promote open science in preclinical imaging. As an example of such an effort, a recent guideline lists ~45 metadata on the use of PDX in cancer research [12]. To support similar activities within the preclinical imaging community, the NCI established the co-clinical imaging research resource program (CIRP) [13]. Among the numerous mandates of the CIRP is to establish a consensus on the requirements of metadata needed to support preclinical imaging research in the era of precision medicine. Similarly to clinical imaging applications, this metadata should enable organization of sustainable database image archives that support queries and computational/statistical analyses.

To this end, the CIRP's imaging informatics and outreach (IMOR) working group (WG) devised a survey to collect metadata needs across a wide range of oncologic preclinical imaging experiment workflows. The survey consisted of nine categories (Table 1) and was disseminated to members of the CIRP network to establish a consensus on the metadata needs for each category. A major consideration in this effort was to reconcile metadata requirements with prior initiatives, including efforts to improve support for small animal imaging according to Digital Imaging and Communications in Medicine (DICOM) [14]. Following several iterations to achieve a consensus, each survey item was deemed as either

"Essential", "Recommended", or "Not needed". It is critical to note that the set of consensus-based metadata should be viewed as fields that should be available to capture the workflow (acquisition, animal model, imaging, etc.) of an oncologic preclinical imaging experiment on the scanner console and/or in an image repository. Not all fields are expected to be used for each experiment. Rather, we anticipate that the fields described within may be further refined based on specific applications—for a given experimental workflow, particular metadata may be required rather than recommended in addition to those deemed required by default. Overall, we anticipate that better image metadata standards will promote open science and enhance reproducibility in oncologic preclinical imaging, and, more broadly, serve as a foundation to expand these principles to other preclinical imaging domains.

**Table 1.** Categories included in the survey.

| Category | Survey Items | Responders |
|---|---|---|
| Animal Identification | 11 | 9 |
| Animal Model | 12 | 9 |
| Animal Feeding | 8 | 9 |
| Environmental/Housing | 52 | 9 |
| Protocol Items | 6 | 9 |
| Imaging Common | 7 | 14 |
| MR Imaging | 109 | 8 |
| PET Imaging | 98 | 4 |
| CT Imaging | 80 | 2 |

## 2. Methods

### 2.1. Design of the Survey

Based on WG discussions, members of the CIRP network [15] created a survey of potential metadata items needed for a range of oncologic animal studies and imaging workflows, including computed tomography (CT), positron emission tomography (PET), single-photon emission tomography (SPECT), and magnetic resonance (MR) preclinical imaging modalities. These items were organized into categories to facilitate the survey. Table 1 tabulates each category, the number of survey questions in each category, and the number of CIRP responders to each category.

The survey was implemented using Google Forms (Alphabet Inc., Mountain View, CA, USA). Each named item in the survey included the corresponding DICOM element name. The survey item name and DICOM element name were the same in many, but not all, cases. A brief text description was included for some items for further clarification. The text description was copied from the DICOM standard where appropriate, or was provided by CIRP members. Survey participants were asked to rate the importance of each item on a five-point scale: "Essential", "Recommended", "Optional", "Unnecessary", and "Unsure". The results were collected, summarized, and given back to the CIRP IMOR WG for review and assessment. Thereafter, the individual items in the survey were rescored using a three-point scale: "Essential", "Recommended", and "Not needed". The score assignments for the metadata items were finalized following discussions within the IMOR WG.

### 2.2. DICOM Viewing Tool

We reviewed a small sample of existing DICOM files from different vendor implementations to test assumptions about the mapping of survey items to DICOM elements. Many software tools exist that dump the metadata from DICOM files, but obtaining a crisp summary normally requires extra manipulation of the output extracted from possibly hundreds of files across multiple imaging studies. Many other open-source tools exist that extract DICOM files for human review and/or machine processing, but these invariably extract the header in a linear format and require the user to find the items of interest. Items that the user wants to review as a group in context are not always adjacent in the DICOM file, thus making it more difficult to collect related metadata for review. To support the

review and analysis of DICOM files, we developed an open-source software tool called DICOM Analysis (https://github.com/Moore-DICOM-Tools/DICOMAnalysis, accessed on 3 May 2023) that supports summary operations and allows the user to organize the output. The DICOM Analysis tool reads one or more DICOM files and generates an output spreadsheet that is controlled by a user-specified profile. At the highest level, the profile defines one or more tabs in the output spreadsheet, where each tab is a logical grouping of metadata. Each tab contains a set of unique values for the items defined in the profile. For example, the summary output might indicate that the administration route for the radiopharmaceutical was encoded in the DICOM metadata in a plain text DICOM element or in a DICOM sequence with a coded value. Further, the output would include only unique values for the administration route, so the user would not need to sort through all the non-unique values extracted from possibly hundreds of DICOM files. Several profiles are included with the software, or the user can write a custom profile. See the Discussion section for more details and use cases.

## 3. Results

The survey results are summarized by category in Tables 2–6 below and in Supplementary Tables S1–S4 with the elements in the associated survey and the WG assessment of the element. Select tables include all survey elements and the associated assessment ("Essential", "Recommended", or "Not needed"). For brevity, the tables for "Environmental—Housing", "Nuclear Imaging", "MR Imaging", and "CT Imaging" contain only elements designated as "Essential" or "Recommended". In some cases, the survey had what might be considered duplicate entries because there were questions about both the text representation and the coded representation of a concept.

**Table 2.** Summary results for the animal identification category.

| Element | Assessment | DICOM Tag | Definition |
|---------|-----------|-----------|------------|
| Patient ID | Essential | 0010,0020 | Primary identifier for the subject. Note: In the case of imaging a group of small animals simultaneously, the single value of this identifier corresponds to the identification of the entire group. |
| Patient's Sex | Essential | 0010,0040 | Sex of the named subject. |
| Patient's Birth Date | Essential | 0010,0030 | Date of birth of the named subject. |
| Patient's Age | Recommended | 0010,1010 | Age of the subject. The DICOM representation for this field includes units (days, weeks, months, and years), and can also be computed from the date of birth and any time point |
| Patient's Weight | Essential | 0010,1030 | Weight of the subject, in kilograms. |
| Patient Species Description | Essential | 0010,2201 | The taxonomic rank value (e.g., genus, subgenus, species, and subspecies) of the patient |
| Strain Description | Essential | 0010,0212 | The strain of the subject. |
| Strain Nomenclature | Essential | 0010,0213 | The nomenclature used for the strain description (0010,0212) |
| Strain Source | Essential | 0010,0218 > 0010,0217 | Identification of the organization that is the source of the animal |
| Strain Stock Number | Essential | 0010,0214 | Strain stock ID at the source. |
| Mouse Strain for Humanized Immune System | Essential | | Background of the mouse strain used for a humanized system (for example, NSG). |
| Type of Humanization | Essential | | Method of humanization—for example, introduction of human CD34+ or PBMC cells. |
| Litter ID | Recommended | | Identification of the mouse's litter. |
| Date Weaned | Recommended | | Date on which the mouse pup was weaned. |

**Table 3.** Summary results for the common animal category.

| Subcategory | Element | Assessment | Comments |
|---|---|---|---|
| PDX | PDX source | Essential | DICOM TID 8182 Exogenous Substance Administration → Brand Name |
| PDX | PDX ID/stock number | Essential | PDX stock number or identification (ID) at the source |
| PDX | Tumor (PDX) passage number | Essential | The passage number of the tumor implanted to generate PDX |
| PDX | Tumor (PDX) passage method | Essential | How tumors were passaged |
| PDX | PDX storage/retrieval/archive Information | Recommended | Method of PDX storage |
| PDX | Tumor implantation method | Essential | Cell suspension or tissue implantation |
| PDX | Number of cells injected if in suspension | Essential | DICOM TID 8182 Exogenous Substance Administration → Usage/Exposure Qualitative Concept |
| PDX | Implant date | Essential | DICOM TID 8182 Exogenous Substance Administration → DateTime Started |
| PDX | Implant site | Essential | DICOM TID 8182 Exogenous Substance Administration → Route of administration → Site of |
| Cell Line | Cell line source | Essential | DICOM TID 8182 Exogenous Substance Administration → Brand Name |
| Cell Line | Cell line ID/Stock number | Essential | Cell line stock number or identification (ID) at the source |
| Cell Line | Cell line name | Essential | DICOM TID 8182 Exogenous Substance Administration → Brand Name |
| Cell Line | Injected site | Essential | DICOM TID 8182 Exogenous Substance Administration → Route of administration → Site of |
| Cell Line | Number of cells injected | Essential | DICOM TID 8182 Exogenous Substance Administration → Usage/Exposure Qualitative Concept |
| GEMM | Mouse name | Essential | Based on International Committee on Standardized Genetic Nomenclature http://www.informatics.jax.org/mgihome/nomen/ (accessed on 3 May 2023) |
| GEMM | Mouse name at source (if different from standard nomenclature) | Essential | Mouse name or reference at source if different from standard nomenclature. |
| GEMM | Source/vendor | Essential | DICOM TID 8182 Exogenous Substance Administration → Brand Name |
| GEMM | Stock/ID number | Essential | Stock number of ID of GEMM at source |

**Table 4.** Summary results for the animal feeding category.

| Element | Assessment | Definition |
|---|---|---|
| Animal Feed | Recommended | The DICOM Animal Feed Type (CID 607) defines 5 coded values with these definitions (codes omitted) or equivalents:<br>• NIH Open Formula Rat and Mouse Ration—18% Crude Protein Autoclavable;<br>• NIH07 open-formula, natural-ingredient rodent diet;<br>• AIN76 purified diet;<br>• AIN93 growth diet;<br>• AIN93 maintenance diet. |
| Feed Source | Recommended | The DICOM Animal Feed Source (CID 608) defines 2 coded values with these definitions (codes omitted):<br>• Commercial product;<br>• Locally manufactured product. |
| Feed Manufacturer | Essential | Free text item in a DICOM structured report. |
| Feed Product Name | Essential | Free text item in a DICOM structured report. |
| Feed Product Code | Recommended | Free text item in a DICOM structured report:<br>• The manufacturer's product code of the feed. |
| Feeding Method | Essential | The DICOM Animal Feeding Method (CID 609) defines 4 coded values with these definitions (codes omitted):<br>• Ad libitum;<br>• Restricted diet;<br>• Food treat;<br>• Gavage. |
| Water Types | Recommended | The DICOM Animal Feeding Method (CID 610) defines coded values with these definitions (codes omitted):<br>• Tap water;<br>• Distilled water;<br>• Reverse osmosis-purified water;<br>• Reverse osmosis-purified, HCl-acidified water. |
| Water Delivery | Essential | The DICOM Animal Feeding Method (CID 609) defines 4 coded values with these definitions (codes omitted):<br>• Ad libitum;<br>• Restricted diet;<br>• Food treat;<br>• Gavage. |

Table 2 and Tables S1–S4 contain columns for the DICOM tag and definition, when available. The absence of a DICOM tag in the table indicates that the field is not available in DICOM, and may potentially need to be added to the small animal DICOM standard. Many of the elements described in these tables will be recorded in DICOM images as acquisition parameters generated by the scanner or other data entered into the console. Not all elements are required by the DICOM standard, and some will not be supported by the scanner manufacturer or may not be entered by the technologist. Still, it is important to provide mappings when available, as well as definitions to clarify the terms. Some elements can be encoded in a DICOM image as free text or as coded values. We included only one of the mappings to clarify this concept. Software implementations will need to determine which encodings are supported by local equipment and collect data accordingly. Tables S1–S4 are discussed below, but are contained in the Supplementary Materials section of this manuscript.

**Table 5.** Summary results for the Environmental—Housing category.

| Element | Assessment | Comments |
|---|---|---|
| Number of Animals Within Same Housing Unit | Recommended | Number |
| Sex of Animals Within Same Housing Unit | Recommended | Code |
| Environmental Temperature | Recommended | Number |
| Housing Humidity (%) | Recommended | Number |
| Heating Conditions | Recommended | DICOM can encode this general topic using the specific items below. |
| Procedure Phase | Essential | Coded values including, but not limited to: preoperative, intraoperative, and postoperative |
| Heating/Heating Method | Essential | The DICOM Heating Method (CID 635) defines 14 values, such as electric blanket, forced air heater, heat lamp, and unheated. |
| Feedback Temperature Regulation | Essential | Yes/No |
| Temperature Sensor Device Component | Recommended | The DICOM Temperature Sensor Device Component Type for Small Animal Procedure (CID 636) defines 3 coded values with these definitions (codes omitted): <br>• Rectal temperature; <br>• Thermography; <br>• Carrier temperature sensor. |

**Table 6.** Summary results for protocol items category.

| Element | Assessment | Comments |
|---|---|---|
| Drug treatment, if relevant | Essential | DICOM Drugs/Contrast Administered (TID 3106) supports the recording of drugs administered in general, including the drug, dose, and route of administration. Medication, substance, and environmental exposure (TID 9002) are more specific to the context of image acquisition. |
| Treatment protocol | Essential | |
| Fasting; fasting duration | Essential | Included in DICOM Imaging Agent Administration Patient Characteristics (TID 10024). Not specific to small animal imaging. |
| Anesthesia used | Essential | DICOM Medication for Small Animal Anesthesia (CID 623) defines 43 coded values. |
| Route of anesthesia administration | Essential | DICOM Anesthesia Induction Code Type for Small Animal Anesthesia (CID 613) defines 5 coded values with these definitions: <br>• Intraperitoneal route; <br>• By inhalation; <br>• Intravenous route; <br>• Per rectum; <br>• Intramuscular route. |
| Chronobiology | Essential | DICOM Circadian Effects (TID 8150) include 3 distinct concepts to express this information: <br>• Total duration of the light–dark cycle (*units = hours*); <br>• Light cycle (the period of time for which a subject is exposed to light, usually expressed as the amount of time in a 24 h cycle; *units = %*) <br>Lights-on time of day (the time of day when the lights are turned on.) |

The majority of values in the DICOM tag column contain a single DICOM Tag represented in the normal GGGG,EEEE format (e.g., 0008,0008) used throughout the DICOM Standard. A small number of values in that column are of the form "GGG1,EEE1 > GGG2,EEE2" (e.g., 0010,0218 > 0010,0217) to indicate that the encoding is contained in a DICOM sequence. Sequences with additional levels are possible in the DICOM standard, but we chose to limit their use in this paper for brevity.

### 3.1. Animal Identification

The items in this category were used to identify the animal experiment subjects. The items include a unique identifier for each rodent as well as attributes describing rodents that are akin to human demographics. Note that the entry for mouse/rodent weight could have been included in the imaging common category, as that value may change over time and/or with any new experiment. Mouse age can be computed from the date of birth and the date of the experiment. The element names and definitions were taken directly from the DICOM standard whenever possible. This was carried out in order to achieve higher specificity, at the cost of introducing what are often terms that describe humans rather than the subjects of small animal imaging.

### 3.2. Oncology Animal Models

Items in the common animal model category describe the characteristics of the animal that is the subject of the experiment. Table 3 includes an extra column for the subcategory. Fields related to PDX were partly derived from a recent publication to identify metadata for PDX models [12].

Table 3 contains items that would not normally be captured at the scanner as part of image acquisition. The DICOM standard anticipates this and provides a mechanism for an information system that stores this information to export these items in a DICOM structured report. Some elements in the other tables can be represented using the DICOM standard, but they are not intended to be included with the image data. These items are intended to be included in DICOM structured reports by an information system to describe aspects of animal handling and not image acquisition. The comments column in Table 3 lists where this information would be encoded using a DICOM structured report.

### 3.3. Animal Feeding, Environment, and Housing

Many quantitative imaging parameters are temperature-dependent; therefore, it was critical that the animals achieved stable core body temperatures and physiological states before the initiation of any quantitative imaging studies [16,17]. Numerous other factors involved in the set-up for preclinical imaging have been documented to impact imaging parameters, including animal handling and diet (duration of fasting), among other factors [16–23]. Parameters related to animal husbandry/housing, including housing conditions, cleanliness, acclimation, chow, strain of the animal, and physiological stress, may also impact the outcome of imaging and therapeutic studies [18]. Imaging studies, typically performed during the day, disrupt the animal's circadian rhythms, which modifies disease metabolism in some cases [24,25]. An important consideration in multi-center preclinical trials is institutional variability in housing. A recent analysis of the data derived from the Mouse Metabolic Phenotyping Centers (MMPCs) study suggests that the "institution" in which a study was performed was a key variable contributing to metabolism (energy expenditure), even when the same diet was used across institutions [26]. Thus, institutional differences in animal housing may also impact preclinical imaging and therapeutic studies. To enhance the reproducibility and the translational impact of preclinical imaging studies, these factors need to be considered and recorded to facilitate the interpretation of co-clinical imaging trials.

Items in the "Animal Feeding" category are summarized in Table 4. The definition column contains the terms defined in the DICOM standard to express these concepts in a DICOM structured report. The DICOM standard allows systems to add values to each list.

This paper does not comment on the value sets to be used in these contexts. The reader is referred to DICOM Part 16 template TID 8122, "Animal feeding", for the details.

The "Environmental—Housing" category included 52 items, most of which were rated as "Not Needed". Table 5, below, will omit those items to make the table more readable. The comments column provides general information for some items and specific encoding for others, all of which are relevant when this information is encoded in a DICOM structured report.

### 3.4. Protocol Items

Protocol items describe the experimental parameters defined by the study. Table 6 summarizes the working group assessments for this category.

### 3.5. Imaging Related Metadata

The imaging-related metadata are provided in the Supplementary Tables due to their length. The metadata are divided into four sections: Imaging Common (Table S1), Nuclear (PET and SPECT) Imaging (Table S2), MR Imaging (Table S3), and CT imaging (Table S4). The "Imaging Common" identified items which were common to all modalities. A total of 4 responders reviewed 98 items in the context of PET and SPECT imaging. Supplementary Table S2 lists only the items rated as "Essential" or "Recommended". A total of 8 responders reviewed 109 items in the context of magnetic resonance imaging. The responders considered the broad range of MR imaging contrasts available including contrast-enhanced imaging, diffusion, arterial spin labeling (ASL), and hyperpolarized $^{13}$C. Table S3 below lists only the items rated as "Essential" or "Recommended". The DICOM MR image definition of Table S3 includes the "Contrast/Bolus", module which assumed a single injection of a contrast agent. Enhanced MR images use the enhanced contrast/bolus module, which supports multiple contrast injections and includes several additional DICOM elements not defined in the contrast/bolus module. Two responders reviewed eighty items in the context of CT imaging. Supplementary Table S4 lists only the items rated as "Essential" or "Recommended". CT imaging is advancing by incorporating multi-energy capabilities to provide spectral data. In preclinical research, most micro-CT scanners still use single-energy scans, with the exception of the MARS preclinical CT scanner from MARS Bioimaging, Ltd., in Christchurch, New Zealand, which uses a photon-counting detector based on the Medipix3 chip from CERN in Geneva, Switzerland. As these systems are rare and still developing, we have chosen not to include multi-energy acquisition/reconstruction in our survey.

### 3.6. DICOM Analysis Tool

A secondary result of this work is the DICOM Analysis tool, which was written to review example preclinical scan datasets. This tool can bolster the user's understanding of how and where manufacturers may choose to encode metadata in DICOM images for small animal imaging. Figures 1 and 2 below show the sample output for the "Imaging Common" and "MR Imaging" sections when the tool was used to extract data from two Bruker MR imaging studies. We believe that this method of organizing data extraction will be useful to investigators who need to review the imaging metadata captured during their experiments.

The column headings are:

- Label: arbitrary string defined by the user of the tool. The user might choose the strings used in the survey, or might choose different strings for a different task;
- DICOM element: the name of the DICOM element (from the DICOM standard);
- DICOM tag(s): one or more tags used to extract data from the DICOM file. Multiple tags are needed when the item is encoded in a DICOM sequence;
- Type (1, 2, 3, etc.): This is the DICOM data element type. The data element type of an attribute of an information object definition or an attribute of a SOP class definition is used to specify whether that attribute is mandatory or optional. The data

element type also indicates whether an attribute is conditional (only mandatory under certain conditions);

- Unique value identified in data: The software uses a simple scheme to report only unique values in this summary output. The tool does not repeat values that have been identified over multiple image sets.

| | A | B | C | D | E |
|---|---|---|---|---|---|
| 1 | Label | DICOM Element | DICOM Tag(s) | Type (1, 2, 3, etc.) | Unique Value Identified in Data |
| 2 | Scanner Unique ID | Device Serial Number | 0018,1000 | 3 | 422180 |
| 3 | Scanner Manufacturer | Manufacturer | 0008,0070 | 2 | Bruker BioSpin MRI GmbH |
| 4 | Scanner Name | Station Name | 0008,1010 | 3 | |
| 5 | Manufacturer's Model Name | Manufacturer's Model Name | 0008,1090 | 3 | BAP94/21 |
| 6 | Software Versions | Software Versions | 0018,1020 | 3 | Acquisition PV-360.2.0.pl.1\ParaVision 360.2.0.pl.1 |
| 7 | Imaging Date | Series Date | 0008,0021 | 3 | 20220430 |
| 8 | Imaging Date | Series Date | 0008,0021 | 3 | 20201116 |
| 9 | Imaging Time | Series Time | 0008,0031 | 3 | 134128 |
| 10 | Imaging Time | Series Time | 0008,0031 | 3 | 105134 |

**Figure 1.** Sample output of DICOM Analysis tool when using the Imaging Common profile.

| | A | B | C | D | E |
|---|---|---|---|---|---|
| 1 | Label | DICOM Element | DICOM Tag(s) | Type (1, 2, 3, etc.) | Unique Value Identified in Data |
| 2 | Field Strength | Magnetic Field Strength | 0018,0087 | 3 | 9.40183463648 |
| 3 | Field Strength | Magnetic Field Strength | 0018,0087 | 3 | 9.40207451955 |
| 4 | Repetition Time | Repetition Time | 0018,0080 | 2C | |
| 5 | Repetition Time | Repetition Time | 0018,0080 | 2C | 886.635724889 |
| 6 | Echo Time | Echo Time | 0018,0081 | 2 | |
| 7 | Echo Time | Echo Time | 0018,0081 | 2 | 8.22612257016 |

**Figure 2.** Sample output of the DICOM Analysis tool when using the MR imaging profile. Many elements are omitted from the output for readability.

The samples below represent unique values from distinct imaging studies, but do not provide the information needed to segregate the results based on the imaging study. Enhancements are being considered for future versions that will allow for segregation and other formatting options.

## 4. Discussion

The work represents consensus opinions from the CIRP IMOR WG. Readers familiar with the DICOM standard know that most of the items mentioned in Tables S1–S4 are commonly encoded in the DICOM headers of clinical images acquired on PET/SPECT, MR, and CT scanners. While the same DICOM requirements apply to images generated from small animal imaging instruments [27], there is a lack of agreement and consistency as to which metadata items are included by the different manufacturers. Designers of informatics platforms for small animal imaging will need to consider that some items that are commonly found in clinical DICOM images may not be readily available in the same DICOM elements, might be found in a text element or coded element, or might not be included at all in small animal systems. Existing preclinical scanners typically store images in a proprietary format and include a module to convert images to DICOM format. Furthermore, the extent of conformance to the DICOM standard varies among preclinical platforms. Even with full compliance, different vendors might make different choices on their preferred encoding using DICOM, making the task of metadata collection and management more difficult. As investigators gain more experience with metadata and the market for small animal image scanners, using the DICOM standard as a primary output, we will find it easier to exchange data for research for both retrospective and ongoing studies. A consensus on preclinical DICOM attributes is the first step to incentivize vendors to standardize their corresponding implementations.

A link between imaging metadata and study-specific attributes is essential to enable the search and selection of data subsets for reproducibility, sharing, and quantitative/statistical analysis. Tables 2–6 identify metadata items that are needed for the study design, but

scanner consoles do not offer the option to capture all of these data. For example, scanner consoles do not accept data for items related to PDX, cell lines, GEMM, or animal feeding, to name a few. Diet and environmental conditions are critical factors that will impact the reproducibility of preclinical studies (imaging and non-imaging) and, thus, need to be captured to qualify an imaging experiment as that detailed in the discussion leading to Tables 4 and 5 (see [16–23]). Provisions to accommodate small animal imaging in the DICOM standard [27] include a mechanism and encodings for some of these metadata elements, but the mechanism is intended to be used by an information system to share data aligned with that system's structure. Complicating the collection of metadata is the use of mouse hotels for imaging multiple animals in a single experiment. The DICOM standard has been extended to support the imaging of multiple mice. However, this must be implemented by scanner manufacturers, and technologists need to enter the data for the separate animals at the console or record it elsewhere. We anticipate that information systems will need to accept DICOM images without these extensions and provide mechanisms for both entering the data related to multiple subjects and splitting the image data into separate objects to facilitate analysis.

The IMOR WG focused on the metadata necessary to support open science and to enhance the reproducibility of preclinical imaging experiments. We anticipate that the fields denoted as "essential" are to be captured for all relevant imaging experiments. Subsequent refinements are needed to classify which recommended metadata are deemed essential for a given experiment. A related effort within the CIRP WGs is the development of preclinical imaging protocols (PIPs) (see article by Gammon et al. [28] in the corresponding Special Issue) and quantitative imaging reports. PIPs provide acquisition details, descriptions, and claims about imaging biomarkers, including the metadata needed to define the experiment. Quantitative image reports can be used to communicate quantitative information extracted from the images using an analysis pipeline. Such information can be captured in multiple formats. One approach is to leverage the capabilities of the DICOM standard structured reporting and apply it to preclinical small animal imaging studies using DICOM-compliant structures that provide the necessary quantitative metadata resulting from the downstream analysis pipeline. Unlike other image formats that may be more convenient for a specific research task, DICOM allows for interoperability between image analysis tools and reusability of analysis results for other purposes, i.e., in accordance with the FAIR (Findable, Accessible, Interoperable, Reusable) principles. To accomplish this, there are initiatives within CIRP, including the use of DICOM objects. While DICOM is well known as a means of transferring and storing medical images, it also supports collecting information on the images. This has the key benefit that additional meta-information on the images (needed for quantitative analysis) can be stored in a DICOM database, along with the images, using a variety of DICOM objects.

To properly describe and qualify acquisition and quantitative experiment measures, it is critical to collect the relevant acquisition and processing metadata for the experiment and to assess the corresponding uncertainties (precision and accuracy) of the proposed quantitative imaging biomarker. For clinical applications, the details of such assessments are summarized in the specifications of the quantitative imaging biomarker alliance (QIBA) profiles [29]. A parallel effort by the CIRP Image Acquisition Data Processing (IADP) WG is focused on the creation of an online repository of PIPs [30] that comprise templates for MR, CT, and PET imaging to capture essential details and parameters for preclinical quantitative imaging pipelines in order to achieve baseline precision of the corresponding quantitative image biomarker measurement. The IADP PIP efforts would benefit from an established consensus on the required metadata for animal models, image acquisition, and processing in order to include them in PIP specifications and standardize their structures for sharing and search ability. For example, to derive reproducible quantitative measures of the apparent diffusion coefficient (ADC) in MR imaging, it is essential that DICOM capture information on the b-values and diffusion-weighted imaging directions used for the diffusion model fit, as well as the scale and units of the quantitative maps generated on the scanners.

Provisions for those attributes are already included in the DICOM standard, but their implementation and adoption is limited. A recent IADP-wide ADC-phantom round-robin (see article by Malyarenko et al. [31] in the corresponding Special Issue) revealed that these essential elements are missing from the current DICOM implementations on animal scanners across multiple sites, leading to (correctable) technical variability in quantitative ADC measurements. In the case of inadequate metadata and/or conformance from vendors, we are also exploring the ability of the research community to develop tools to manipulate the DICOM metadata. This would be in the form of a program that could be run after data acquisition to update or add metadata to DICOM files. So far, we have found that PyDicom [32], an open-source Python package for working with DICOM files, can support such tools.

An unmet need to support the collection of the metadata, described within, is an informatics platform intended not only to support the collection and maintenance of the metadata, but also to provide tools to query and analyze image datasets. Legacy preclinical imaging databases are not equipped to support big data science or the collection of metadata/annotations to support NCI's precision medicine initiative. While some institutions have developed database platforms to house preclinical imaging datasets, many such legacy databases are not compatible with the complexity and growing demands of preclinical cancer imaging, which include big data needs, collection of metadata/annotation, and integration with the clinical arm of co-clinical trials. Importantly, the increasing prevalence of quantitative acquisition and analysis pipelines depend on sophisticated computational methods that generate additional derived data. Given these big data challenges, informatics tools are needed to organize data structures, enforce quality assurance practices, generate audit trails and provenance records, provide detailed reports and data tracking tools, and ultimately facilitate data analysis. To address the need for a preclinical informatics/database platform, several efforts have been reported, including the Web-Based Application for Biomedical Image Registry, Analysis, and Translation (BiRAT) [33]; Small Animal Shanoir (SAS) [34]; Small Animal Big Data Warehouse Environment for Research (SABER) [35]; and XNAT-PIC [36]. The latter is based on XNAT [37], an open-source imaging informatics platform originally developed to support neuroinformatics work that has recently been extended to support cancer research [37,38]. The VivoQuant PACS platform and Flywheel cloud-based image repository offer potential commercial solutions. A recently introduced NCI Imaging Data Commons (IDC) repository, which includes a number of preclinical collections, utilizes the DICOM standard to harmonize data representation and access and co-locates the data with the tools for search, visualization, and analysis [39].

While all of the aforementioned informatics platforms provide capabilities to support preclinical image archival at various levels, none fully support the complex collection of metadata, annotation needs, and computational pipelines that generate additional derived data in preclinical imaging research. A recent effort supported by NCI's Information Technology for Cancer Research (ITCR), and in collaboration with the CIRP network, aims to develop an open-source preclinical imaging XNAT-enabled informatics (PIXI) platform (https://www.pixi.org/, accessed on 3 May 2023) to support the growing needs of the preclinical oncologic imaging community. Specifically, PIXI aims to manage the workflow of multimodal preclinical imaging studies; harmonize preclinical imaging databases with imaging-associated experiments, including metadata and annotations; and develop and enable the deployment of analytic and computational pipelines in the cloud to support quantitative preclinical imaging research and integration with the clinical components. PIXI is based on XNAT, and will provide an open-source solution that investigators can install in labs/imaging facilities to manage and operate on their data using a centralized system in a federated framework. The results of the CIMI survey will directly inform PIXI metadata needs and image database organization. An initial software release is planned for the fourth quarter of 2023, which will allow early adopters to provide feedback to the development team.

## 5. Conclusions and Future Directions

We describe a CIRP consensus of preclinical imaging metadata needs across a range of oncologic preclinical imaging experiment workflows. This effort was reconciled with prior efforts to enhance the DICOM standard in order to support preclinical imaging. The metadata should be viewed as fields that are available to capture the workflow of an oncologic preclinical imaging experiment. We anticipate that the metadata described herein may be further refined based on specific applications, in that for a given experimental workflow, select metadata may be required, rather than recommended, in addition to those deemed essential by default.

The workflows described herein are not exhaustive. Additional effort is needed to add metadata for preclinical imaging instruments and specialized workflows (such as C-13 MR spectroscopy) not fully described here. Importantly, efforts within the imaging community and industry are needed to update the DICOM standard to add metadata which are not currently available, to define dictionaries and ontologies, and to harmonize the mapping of metadata in the DICOM standard across instrument manufacturers. A critical unmet need is an informatics platform which will not only support the collection and maintenance of the metadata, but also provide tools for queries and analyses. Several efforts are underway to that end, including through academic sites and industry. Overall, we anticipate that better reporting standards will promote open science, standardization, and reproducibility in preclinical imaging in general, as well as facilitate integration with the clinical arms of co-clinical trials.

**Supplementary Materials:** The following supporting information can be downloaded at: https://www.mdpi.com/article/10.3390/tomography9030081/s1. Tables S1–S4 list elements of four different survey categories. The DICOM elements in the tables are organized according to the modules defined in the DICOM standard to simplify review by those who are familiar with that organization. Table S1 includes elements common to all imaging modalities. The elements for PET and SPECT data in Table S2 include DICOM elements that can be found in the DICOM positron emission tomography image specification. The elements for MR data in Table S3 include DICOM elements that can be found in the DICOM MR image or DICOM enhanced MR image specifications. The elements for CT data in Table S4 include DICOM elements that can be found in the DICOM CT image or DICOM enhanced CT image specifications. Reference [40] is cited in the Supplementary Materials.

**Author Contributions:** Writing (original draft), S.M.M. and K.I.S.; methodology, S.M.M.; software, S.M.M. and G.D.A.; validation, S.M.M., J.D.Q., A.W.L., G.D.A., R.L. and C.T.B.; editing and reviewing, J.D.Q., R.L., A.W.L., P.E.Z.L., R.S., J.K., C.T.B., M.H., A.Y.F., P.E.K., A.M.H., T.L.C., D.M., B.D.R., S.P., J.C.G., R.Z., S.T.G., H.C.M., R.R., A.M.H., H.E.D.-L., M.T.L., D.L.R. and T.E.Y.; conceptualization and supervision, K.I.S.; funding acquisition, K.I.S., C.T.B., H.C.M., R.Z., B.D.R., M.T.L., J.K., P.E.K. and H.E.D.-L. All authors have read and agreed to the published version of the manuscript.

**Funding:** The work was supported by the following awards from NCI's CIRP and ITCR programs: U24CA253531, U24CA209837, U24CA264044, U24CA237683, U24CA220245, U24CA264298, U24CA253377, U24CA231858, U24CA220325, U24CA226110, and P50CA228944.

**Institutional Review Board Statement:** Not applicable.

**Informed Consent Statement:** Not applicable.

**Data Availability Statement:** All data generated by the CIRP network are available at each CIRP site's web resource website. The CIRP research resource websites are available at: https://ncihub.cancer.gov/groups/cirphub/resources_page (accessed on 3 May 2023).

**Acknowledgments:** The authors wish to thank the National Cancer Institute's (NCI) Information Technology for Cancer Research (ITCR) and the Co-Clinical Imaging Research Resource Program (CIRP) for supporting the work.

**Conflicts of Interest:** M.T.L. is a founder of and limited partner in StemMed Ltd., and a manager in StemMed Holdings, LP, its general partner. He is also a founder of and equity stake holder in Tvardi Therapeutics Inc.

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
