# Peer review of "Co-Clinical Imaging Metadata Information (CIMI) for Cancer Research to Promote Open Science, Standardization, and Reproducibility in Preclinical Imaging"

_tomography, doi:10.3390/tomography9030081_

Round 1

Reviewer 1 Report

This article represents an important position statement to promote open science and standardisation in pre-clinical imaging.  I have some minor issues for the authors to consider:

1) Whilst the concept of "co-clinical imaging trials" is recognised, I think the terminology "co-clinical models" used in the abstract, and towards the end of paragraph 1 of the Introduction, is inappropriate and actually confusing, especially to the cancer research community.  I think it less ambiguous and better to just refer to PDX, GEMMSs throughout.  On this theme, there is also continued use of orthotopic and ectopic models in pre-clinical imaging-embedded investigations which can be equally valuable and informative (e.g. use of murine tumor models in syngeneic hosts for evaluating response to immunotherapies) which the authors seemingly ignore.

2) With regard to assessing imaging reproducibility, did the authors consider pre-clinical investigations where a double-baseline acquisition was performed prior to intervention?  Whilst not commonly performed in pre-clinical studies (owing to the typically rapid tumor growth), there are some reports that have incorporated such study designs that have enabled the reporting of repeatability of imaging biomarkers.

3) Reporting of the first four categories in the survey listed in Table 1 (Animal identification, animal model, animal feeding and environmental/housing) was originally recommended in the Animal Research: Reporting of In Vivo Experiments or ARRIVE guidelines (Kilkenny et al, PLoS Biol, 2010) and recently updated (ARRIVE 2, du Sert et al, PLoS Biol 2020).  The ARRIVE guidelines are being increasingly adopted for pre-clinical research and their use should be recognised/acknowledged here.  Other well-recognised and related guidelines (e.g. the UKCCCR Guidelines for the Welfare of Animals in Experimental Neoplasia, Workman et al, Br J Cancer 1998), which also covers in vivo imaging, may also be appropriate to acknowledge here.

4) For the Cell Line entries in the Common Animal category results in Table 3, would it not be essential to provide information on the source of the cell line, the vitro passage number of the cells injected, and the date of the most recent STR profiling of the cell line, to provide confidence in, and ensure the provenance of the type of cells/tumors under investigation.

5) In Table 4, are the animal feeds listed only available in the USA?  If so, perhaps the words "or equivalent" should be added.

6) In Table 5, replace "Housing Unit" with "Housing Cage".

Author Response

Comment: Reference to co-clinical models: The reviewer felt the term co-clinical models “is inappropriate and actually confusing, especially to the cancer research community” and suggested use of PDX, GEMMSs throughout.

Response:  We changed reference to co-clinical models to PDX and GEMMs as suggested.

Comment: The reviewer also noted that “continued use of orthotopic and ectopic models in pre-clinical imaging-embedded investigations which can be equally valuable and informative (e.g. use of murine tumor models in syngeneic hosts for evaluating response to immunotherapies)” needs to be considered.

Response:  We thank the reviewer for the comment.  The proposed metadata can indeed identify syngeneic hosts, and other applications.  For example, for syngeneic hosts, the tumor cell line needs to match the mouse species.  The metadata describing the mouse model (species, background, etc.) and the identifying metadata about the cell line can be used to that end.   Thus, we believe that the fields described within will accommodate syngeneic mouse models.  

Comment: With regard to assessing imaging reproducibility, did the authors consider pre-clinical investigations where a double-baseline acquisition was performed prior to intervention?

Response:  We believe the reviewer is referring to test/retest studies to characterize the repeatability of the imaging biomarker.  Claims about repeatability are beyond the scope of the manuscript.  A separate effort initiated by the CIRP network addresses claims about repeatability of an imaging biomarker.  This has been addressed in the discussion section in the discussion about preclinical imaging protocols (PIPs).

Comment: The reviewer suggested referencing relevant prior guidelines provided by ARRIVE and UKCCCR in animal studies and in vivo experiments.

Response:  We appreciate the reviewer’s comment.  These guidelines have been added to the manuscript (see Introduction section).

Comment: For the Cell Line entries in the Common Animal category results in Table 3, would it not be essential to provide information on the source of the cell line, the vitro passage number of the cells injected, and the date of the most recent STR profiling of the cell line, to provide confidence in, and ensure the provenance of the type of cells/tumors under investigation.

Response:  We agree with the reviewer in that source and ID would be essential.  This has been an omission from the table and has been added.  As it relates to STR profiling, this is likely beyond the scope of the imaging metadata.

Comment: In Table 4, are the animal feeds listed only available in the USA? If so, perhaps the words "or equivalent" should be added.

Response:  Thank you for the suggestion.  “Or equivalent” has been added in Table 4.

  1. Comment: In Table 5, replace "Housing Unit" with "Housing Cage".

Response: Thank you for the suggestion. The small animal DICOM refers to “Housing Unit.” For this reason, we opted for “Housing Unit.”

Reviewer 2 Report

This is an important and timely article - Standardization of pre-clinical imaging data acquisition will be essential in order that data can be shared and analyzed with confidence. No major concerns.

Comments (not concerns)

Although the work was conducted by a co-clinical imaging working group, their suggestions will apply broadly to pre-clinical imaging in general.

The lengthy table of MRI metadata is especially important due to the plethora of types of MRI studies and different protocols that can be employed. Standardization, or at least standardized descriptions of how studies are conducted, will be essential in order to compare findings across studies and institutions.

The author have rightfully focused on imaging metadata. However I fear there are additional blocks of data that also need to be captured beyond the scope of the present work, in order to enhance robustness and reproducibility. For example, in the nuclear field there is a huge spectrum of novel, home-grown radiotracers (of different origins and quality). Hopefully more and more will become standardized and commercially available. Another major biological concern in the immunological status of the animals. There can be subtle differences in how "clean" animal housing can be, from source to source, facility to facility, and institution to institution. When conducting cancer immunotherapy studies, this can result in great variations in tumor take and responses to therapy.  Of course this is well beyond the scope of the current work but the authors might want to discuss and highlight other areas where standardization and capture of metadata will be important to tackle as well.

The authors are all from academia. Clearly, involvement of industry would be helpful, perhaps at the next stage.

The description of existing and emerging repositories is especially timely given the new NIH Data Management and Sharing Policy. Investigators will need places to house large image databases - and in order for sharing to be beneficial, standardization and inclusion of more metadata will be essential.

Minor comment

p 14, first full paragraph, first sentence. "archival" should be "archiving"? regardless, it needs to be a noun.

Author Response

Comment: “…The author have rightfully focused on imaging metadata. However, I fear there are additional blocks of data that also need to be captured beyond the scope of the present work, in order to enhance robustness and reproducibility. For example, in the nuclear field there is a huge spectrum of novel, home-grown radiotracers (of different origins and quality)…”

Response:  The reviewer is likely referring to quality control/quality assurance of a tracer as it related to metabolites, etc.  The reviewer is correct in that QA/QC of a tracer is beyond the scope of DICOM reference presented within.  However, an informatics databases can capture such information, and this will be addressed through the development of databases described in the discussion section.

Comment: “…Another major biological concern in the immunological status of the animals…”

Response:  Immunological status of animals is defined in Table 2.  “Strain Nomenclature” captures any genetic modifications.  In addition, two additional fields capture humanization of mouse models.

Comment: “There can be subtle differences in how "clean" animal housing can be, from source to source, facility to facility, and institution to institution. When conducting cancer immunotherapy studies, this can result in great variations in tumor take and responses to therapy. Of course, this is well beyond the scope of the current work but the authors might want to discuss and highlight other areas where standardization and capture of metadata will be important to tackle as well…”

Response:  Thank you for the suggestion.  As the reviewer noted, a full discussion is beyond the scope of the manuscript, however, in the results section under the heading “Animal Feeding, environment, and housing” we note the role of housing (husbandry) on imaging and therapeutic outcome.

Comment: “The authors are all from academia. Clearly, involvement of industry would be helpful, perhaps at the next stage.”

Response:  Thank you for the comment.  The effort materialized out of an academic support to support co-clinical imaging research.  Once the document is published, we intend to include industry in the discussion through engagement in scientific meetings (such as the annual CIRP meeting).  This is a starting document, and hopefully future efforts will involve broad industry contribution.

Comment: first full paragraph, first sentence. "archival" should be "archiving"? regardless, it needs to be a noun.

Response:  We cannot locate the reference to “archival” in the first full paragraph, first sentence. 

Reviewer 3 Report

General comments: Well written manuscript. Timely topic.

Specific comments:

Title: appropriate

Abstract: no changes recommended

Introduction/body/conclusion: adequate

Tables/figures: adequate

References: adequate

The manuscript is a bit boring and long.  It would benefit by editing to make it briefer. Potentially, some of the tables could be moved into a supplementary section.

Author Response

Comment: Reviewer #3 suggested that given the length of the manuscript, some of the tables could be moved as supplemental figures. 

Response:  We understand and agree with the reviewer that the manuscript is long.  We have moved several of the longer tables to the supplemental section.  The tables that remain in the main body support the flow of the manuscript to minimize back and forth and ease of reference.